# Genome-Wide Identification and Analysis of the *BBX* Gene Family and Its Role in Carotenoid Biosynthesis in Wolfberry (*Lycium barbarum* L.)

**DOI:** 10.3390/ijms23158440

**Published:** 2022-07-29

**Authors:** Yue Yin, Hongyan Shi, Jia Mi, Xiaoya Qin, Jianhua Zhao, Dekai Zhang, Cong Guo, Xinru He, Wei An, Youlong Cao, Jianhua Zhu, Xiangqiang Zhan

**Affiliations:** 1State Key Laboratory of Crop Stress Biology for Arid Areas, College of Horticulture, Northwest A&F University, Yangling, Xianyang 712100, China; yinyue2011@nwafu.edu.cn (Y.Y.); shihongyan@nwafu.edu.cn (H.S.); zdk11@nwafu.edu.cn (D.Z.); guo_cong@nwafu.edu.cn (C.G.); 2National Wolfberry Engineering Research Center, Ningxia Academy of Agriculture and Forestry Sciences, Yinchuan 751002, China; lorna0102@126.com (J.M.); qinxiaoya@whu.edu.cn (X.Q.); zhaojianhua0943@163.com (J.Z.); hhexinru@163.com (X.H.); 13027986722@163.com (W.A.); caoyoulong@nwberc.com.cn (Y.C.); 3Department of Plant Science and Landscape Architecture, University of Maryland, College Park, MD 20742, USA

**Keywords:** *Lycium barbarum*, *LbaBBX* gene family, *LbaBBX* gene expression, carotenoid biosynthesis, protein subcellular localization

## Abstract

The B-box proteins (BBXs) are a family of zinc-finger transcription factors with one/two B-Box domain(s) and play important roles in plant growth and development as well as stress responses. Wolfberry (*Lycium barbarum* L.) is an important traditional medicinal and food supplement in China, and its genome has recently been released. However, comprehensive studies of *BBX* genes in *Lycium* species are lacking. In this study, 28 *LbaBBX* genes were identified and classified into five clades by a phylogeny analysis with BBX proteins from *Arabidopsis thaliana* and the *LbaBBXs* have similar protein motifs and gene structures. Promoter *cis*-regulatory element prediction revealed that *LbaBBXs* might be highly responsive to light, phytohormone, and stress conditions. A synteny analysis indicated that 23, 20, 8, and 5 *LbaBBX* genes were orthologous to *Solanum lycopersicum*, *Solanum melongena*, *Capsicum annuum*, and *Arabidopsis thaliana*, respectively. The gene pairs encoding LbaBBX proteins evolved under strong purifying selection. In addition, the carotenoid content and expression patterns of selected *LbaBBX* genes were analyzed. *LbaBBX2* and *LbaBBX4* might play key roles in the regulation of zeaxanthin and antheraxanthin biosynthesis. Overall, this study improves our understanding of *LbaBBX* gene family characteristics and identifies genes involved in the regulation of carotenoid biosynthesis in wolfberry.

## 1. Introduction

Zinc finger transcription factors (TFs) are some of the most abundant TFs in plants and play a vital regulatory role in the regulation of transcription and various biological functions [1,2]. B-Box (BBX) proteins are a class of zinc-finger TFs possessing one or two B-box domains (CX2CX8CX7CX2CX4HX8H) in the N-terminus; some have an additional CCT (CONSTANS, CO-like, and TOC1) conserved domain or VP (valine–proline) motifs in the C-terminus. The B-box domains can be classified into two types: B-box1(B1) and B-box2 (B2). Two B-box conserved domains are recognized based on their consensus sequence and the distance between the zinc-binding residues [3]. Potential segmented duplication and deletion events result in differences in the consensus sequences and space between the zinc-binding residues in the two B-box domains [3,4]. In addition, the highly conserved CCT domain is comprised of 42–43 amino acids and is important for the regulation of transcription and nuclear protein transport [5,6]. According to the existence of BBX and CCT domains, 32 BBX proteins have been identified and classified into five subgroups in *Arabidopsis* [7]. Therefore, members of BBX proteins are divided into five categories depending on the presence of B-Box domains along with the CCT domain and have been reported in multiple species [4].

Subsequently, many studies have shown that plant BBX proteins play important roles in diverse physiological and biochemical processes, such as flowering time regulation, photomorphogenesis, shade avoidance, secondary metabolism, and biotic and abiotic stress responses [8,9,10,11,12]. The first *BBX* gene (*CONSTANS (CO)*, known as *AtBBX1*) was identified and characterized in *Arabidopsis*; it can activate FLOWERING LOCUS T (FT) by binding to its promoter under a long day length [13]. Other *BBX* genes were subsequently discovered and functionally characterized, including *BBX4*, *BBX6*, *BBX7*, and *BBX3**2*, with roles in the regulation of flowering time [14,15,16]. Recently, BBX proteins have been reported to regulate secondary metabolism in fruits, especially anthocyanin and carotenoid biosynthesis. In *Arabidopsis*, BBX21/22/23 are positive regulators of anthocyanin accumulation [10], while BBX24/25/31 negatively regulate anthocyanin biosynthesis in response to several environmental factors. The overexpression of *VvBBX44* decreased the expression of *VvHY5* and *VvUFGT* and reduced the anthocyanin content in grape calli [17]. In pear, *PpBBX16,* a homolog of *AtBBX22*, is a positive regulator of light-induced anthocyanin accumulation [18]. In apple, MdBBX1/20/21/22 and MdBBX33 promote anthocyanin biosynthesis by light-induced anthocyanin accumulation, whereas MdBBX37 is a negative regulator of anthocyanin accumulation via light signaling. In tomatoes, *SlBBX20* regulates the synthesis of carotenoids by directly binding to the promoter of the gene encoding the carotenoid biosynthesis enzyme PHYTOENESYNTHASE 1 [11]. However, studies of *BBX* genes in wolfberry are rare. 

Wolfberry (*Lycium barbarum* L.; 2n = 2x = 24), a fruit tree in the family of Solanaceae, is an important medicinal and edible plant in China. *L. barbarum* is a rich source of carotenoid esters, which are mainly composed of zeaxanthin dipalmitate, lutein palmitate, antheraxanthin, and β-cryptoxanthin. Therefore, carotenoids are responsible for the orange, yellow, and red colors of *L. barbarum* fruits [19,20]. During the past few decades, many studies have identified TFs families with important roles in the regulation of carotenoid biosyntheses, such as *SlMYB72*, *SlWRKY35*, *MdAP2*, and *SlBBX20* [11,21,22,23]. Two R2R3-MYB family members, *LbaMYB26*(*Lba02g01219*) and *LbaMYB123* (*Lba11g01830*), are candidate genes involved in the regulation of carotenoid biosynthesis in *L. barbarum* fruits [24]. The *BBX* gene family has been identified and evaluated in many plant species, such as *Solanum lycopersicum* [25], *Capsicum annuum* [26], *Iris germanica* [27], *Prunus avium* [28], *Vitis Vinifera* [29], and *Arabidopsis thaliana* [30], and has diverse functions. Our understanding of the functions of the *BBX* gene family, such as roles in responses to biotic and abiotic stresses and secondary metabolite biosynthesis, has advanced. However, the mechanism by which *BBX* genes contribute to the regulation of carotenoid biosynthesis is still unclear.

Comprehensive studies of *BBX* genes in wolfberry have not been reported to date. The recent completion of the *L. barbarum* genome provides a basis for investigating the *BBX* gene family in the species at the genome level [31]. To further characterize the *BBX* gene family in *Lycium*, we performed systematic genome-wide identification and analyses of the *BBX* gene family, bridging the research gap in *BBX* gene family studies. Analyses of physical and chemical characteristics, collinearity analysis, phylogenetic and evolutionary relationships, conserved domains, gene structures, cis-regulatory networks, subcellular localizations, and expression patterns of *LbaBBX* genes were performed. This study lays a foundation for further analyses of the roles of *LbaBBX* genes in carotenoid biosynthesis and fruit development in wolfberry.

## 2. Results

### 2.1. Identification and Characteristics of LbaBBX Genes 

To identify *BBX* genes in the wolfberry genome, hidden Markov model (HMM) searches with the B-box domain HMM profile (PF00643) and BLSATP using 32 BBX protein sequences from *Arabidopsis thaliana* as queries were performed. The candidate BBX protein sequences were used to detect the presence of B-box conserved domains by the Simple Modular Architecture Research Tool (SMART) and the National Center for Biotechnology Information (NCBI) batch CD-Search. A total of 28 putative *LbaBBX* genes were identified (Table 1). These *BBX* genes were named *LbaBBX1* to *LbaBBX28* according to their location on the *L. barbarum* chromosomes. The coding sequences (CDS) of *BBX* genes ranged from 330 bp to 1374 bp. They encoded proteins that were 109 to 457 amino acids (AA) in length, with predicted putative molecular weights ranging from 12.49 kDa to 51.73 KDa. The grand average of hydropathicity (GRAVY) values for all BBXs were negative, indicating that the BBX proteins were hydrophilic. The subcellular localization results showed that most of the LbaBBX proteins were found in the nucleus.

### 2.2. Protein Domains and Phylogenetic Analysis of LbaBBX Proteins

The conserved sequences of B-Box domains (B-Box1 and B-Box2), CCT domain, and VP motif in wolfberry BBX proteins were identified, and sequence logos are shown in Appendix A. Out of 28 LbaBBXs, ten contained two B-box domains and a conserved CCT domain, whereas three members had a valine–proline (VP) motif. Three and six members contained one B-box domain plus a CCT domain and only one B-Box domain, respectively, and the remaining nine members contained two B-Box domains. Conserved structures of LbaBBX members were found with B-box1 sequence (C-X2-C-X8-C-X2-D-X4-C-X2-C-D-X3-H-X8-H-X-R-X, X represents any amino acid) and B-box2 (C-X2-X8-C-X8-C-C-X3-X9-H-X-R-X4). Additionally, the CCT domain was highly conserved. Multiple sequence alignments of B-box1, B-box2, CCT domain, and VP motif for all LbaBBX proteins were also generated (Appendix A). Based on the alignments, some absolutely conserved amino acid residues were found, such as the Cysteine (C) and Histidine (H) residues in the B-box domain, Arginine (R) and Lysine (K) residues in the CCT domain, and Valine (V) and Proline (P) residues in the VP motif. 

The full-length amino acid sequences were used to construct a phylogenetic tree by the maximum likelihood (ML) method using IQ-TREE. As shown in Figure 1a, the LbaBBX family was divided into five subgroups, consistent with previous studies of the gene family in tomato, pepper, and *Arabidopsis* [25,26,30]. We found that LbaBBX proteins assigned to the same group possess similar domain organizations. For example, in subgroup I, three LbaBBXs contained two B-Box domains, an additional CCT domain, and a VP domain (Figure 1a). In order to confirm the subfamily clustering of BBX members in wolfberry, a phylogenetic tree of LbaBBX together with SlBBX, CaBBX, StBBX, SmBBX, IcBBX, and AtBBX was also constructed by using the ML method (Appendix A). All BBX proteins were also divided into five subfamilies. Furthermore, the sequences of B-box 1 (Figure 1b), B-box 2 (Figure 1c), and CCT (Figure 1d) domains were also evaluated. The members of subgroups I and II contained both B-Box and CCT domains, except for LbaBBX23, which harbored only two B-Box domains. Members of subgroup III had one B-box domain and one CCT domain. Members of subgroups IV and V had no CCT domain and only two or one B-Box domain(s), respectively. 

### 2.3. Gene Structure and Motif Composition of the LbaBBX Gene Family

The exon–intron structures and conserved motifs were examined to gain insight into the structural diversity of *LbaBBX* genes. As shown in Figure 2b, the number of exons ranged from one to five, with an average of 2.9. Additionally, wolfberry *BBX* genes in clades I, II, III, and Ⅳ exhibited highly similar gene structures; however, *LbaBBX* genes in clade V showed highly variable structures. For example, most of the *LbaBBX* genes in clades I, II, III, and Ⅳ possessed two, four (except *LbaBBX23*), two, and four (except *LbaBBX16*) genes, respectively (Figure 2a). These results suggested that exon losses or gains occurred during the evolution of the gene family and resulted in functional divergence among closely related *LbaBBXs.*


To further examine the structural features of LbaBBX proteins, the conserved motif compositions were analyzed using MEME. Fifteen conserved motifs were predicted and named motifs 1–15 (Figure 2c). Motifs one and four were found in all LbaBBX proteins except LbaBBX3. Most of the *LbaBBX* genes assigned to the same group shared similar motif compositions and arrangements, which further validated the classification results. For example, motifs 6 and 13 were detected only in groups II and I, respectively. Three LbaBBX members (LbaBBX9, LbaBBX10, and LbaBBX12) from group II possessed maximum motifs, containing motifs 1, 2, 3, 4, 6, 8, and 14. Except for LbaBBX7, LbaBBX26 and LbaBBX28 harbored only two motifs (motif one and motif four) in group V. The detailed sequence information for these 15 motifs is shown in Appendix A.

### 2.4. Chromosomal Location and Duplication of LbaBBX Genes

We plotted the *LbaBBX* genes on the chromosomes of the wolfberry genome (Figure 3a). A total of 28 *LbaBBX* genes were evenly distributed on 11 of 12 wolfberry chromosomes, and the number of *LbaBBX* genes on each chromosome was not related to the chromosome size (Appendix A). Each *LbaBBX* gene name corresponds to its physical position from the top to the bottom of *L. barbarum* chromosome 1 to chromosome 12. Chromosome 4 contained the largest number of *LbaBBX* genes (6 genes, ~21.4%), followed by chromosome 5 (5 genes, ~17.9%) and chromosome 11 (5 genes, ~17.9%). Only one *LbaBBX* gene was located on each of chromosomes 1, 2, 3, and 12, and only two were detected on chromosomes 6, 9, and 10. No *LbaBBX* genes were located on chromosome 8.

Different patterns of gene duplication contributed to the evolution of the *BBX* gene family, including whole-genome duplication (WGD) as well as segmental duplication, tandem duplication (TD), proximal duplication (PD), transposed duplication (TRD), and dispersed duplication (DSD). We used DupGen_finder [32] to detect duplicated *BBX* family gene pairs in wolfberry. The numbers of DSD, WGD, TRD, PD, and TD duplication events in wolfberry were 23, 12, 4, 1, and 1, respectively (Figure 3b). These results indicated that DSDs and WGDs explained the majority of gene duplication events in the *LbaBBX* gene family.

In addition, collinearity was analyzed among homologous regions in other species, including *Solanum lycopersicum*, *Capsicum annuum*, *Solanum melongena*, and *Arabidopsis thaliana.* The interspecific collinearity results revealed 56 orthologous pairs (Figure 3c). Orthologous relationships were detected between *LbaBBX* genes and genes in four species belonging to Solanaceae and *A. thaliana*, including *L. barbarum*–*S. lycopersicum* (23 pairs), *L. barbarum*–*S. melongena* (20 pairs), *L. barbarum*–*C. annuum* (8 pairs), and *L. barbarum*–*A. thaliana* (5 pairs) (Appendix A). The numbers of orthologous events of *LbaBBX-SlBBX*, *LbaBBX-SmeBBX*, and *LbaBBX-CaBBX* were greater than that of *LbaBBX-AtBBX*. These results indicated that wolfberry was closely related to the other three species in Solanaceae. The high numbers of orthologous events of *LbaBBX-SlBBX* identified in our study suggest that *LbaBBX* genes in wolfberry share a similar structure and function to those of *SlBBX* genes in tomato.

### 2.5. Nonsynonymous (Ka) and Synonymous (Ks) Substitutions per Site, and Ka/Ks Analysis of BBX Family Genes

We estimated rates of synonymous (*K*_s_) and nonsynonymous (*K*_a_) substitutions for 56 duplicated gene pairs. As illustrated in Figure 4, the *K*_a_/*K*_s_ values for WGD-derived gene pairs in wolfberry ranged from 0.172 to 0.403, and the *K*_a_/*K*_s_ values for gene pairs derived from DSD, TRD, PD, and TD were 0.098–0.527, 0.199–0.227, 0.357–0.357, and 0.444–0.444, respectively (Figure 4 and Appendix A). In general, *K*_a_/*K*_s_ value greater than 1.0 provide evidence for positive selection, values less than 1.0 suggest purifying selection, and values equal to 1.0 suggest neutral evolution. In our study, all *LbaBBX* gene pairs had *K*_a_/*K*_s_ values less than 1, indicating that these genes primarily underwent strong purifying selection.

### 2.6. Cis-Regulatory Elements in the Promoters of LbaBBX Genes

The 1500 bp upstream sequences of the 28 *LbaBBX* genes were extracted for analyses of the *cis*-regulatory elements in the promoter regions. In total, 482 *cis*-acting elements were identified and classified into three basic categories, including plant growth and development, phytohormone, and stress responses (abiotic/biotic) (Figure 5 and Appendix A). In the first subgroup (i.e., plant growth and development), the majority (84.5%) of elements were light-responsive elements, such as GT1-motif, Box 4, G-box, and I-box, which are widespread in plants (Figure 5c). The second subgroup included elements involved in phytohormone responses; the ABREs for abscisic acid (ABA) responsiveness were the most common elements, appearing 55 times in 28 *LbaBBXs*, accounting for 27.9% of the hormone-responsive *cis*-regulatory elements (Figure 5c). The others were the CGTCA-motif and TGACG-motif for MeJA-responsiveness elements, TATC-box and P-box for gibberellin-responsive elements, and TGA-element for auxin-responsive elements, suggesting that LbaBBXs are regulated by various hormones (Figure 5c). The last subgroup included elements related to different stress responses. A cis-acting regulatory element essential for anaerobic induction (ARE) was identified in 19 *LbaBBX* gene promoters (Appendix A), suggesting that these genes might be induced by low oxygen levels. W box (wounding and pathogen responsiveness), LTR (low-temperature responsiveness), and MBS (MYB binding site involved in drought-inducibility) were also found. Furthermore, MYC was found in 24 *LbaBBX* genes, suggesting that *LbaBBXs* contribute to the response to abiotic stress (Appendix A).

### 2.7. Expression Patterns of LbaBBX Genes in Different Tissues

To further understand the dynamic gene expression patterns of *BBX* gene family members in *L. barbarum*, we evaluated expression profiles in four tissues (leaf, stem, flower, and fruit) with RNA-seq analysis. The *LbaBBX* genes exhibited tissue-specific expression and were further divided into three groups (Figure 6 and Appendix A). In Group I, six genes (*LbaBBX16*, *LbaBBX25*, *LbaBBX1*, *LbaBBX19*, *LbaBBX20*, and *LbaBBX21*) presented high overall expression levels in all four organs, suggesting that these *LbaBBX* genes play important roles in the formation of these tissues, except two genes (*LbaBBX15* and *LbaBBX25*) in leaves with relatively low expression. Of 28 genes, seven BBXs were assigned to Group II. Remarkably, not all homologous gene pairs exhibited similar patterns of expression; for example, *LbaBBX9* had the highest transcript abundance in the leaf, and *LbaBBX26* expression was highest in the stem (Figure 6b). In Group III, the remaining 14 genes shared similar low expression levels in these tissues, except *LbaBBX4* (which was highly expressed in fruits). Additionally, several genes that were highly expressed in the fruit were identified, including *LbaBBX1*, *LbaBBX4*, *LbaBBX16*, and *LbaBBX25* (Figure 6b). 

### 2.8. Identification LbaBBX Genes Related to Carotenoid Contents

In order to ascertain how *LbaBBX* gene expression (FPKM values) may be predictive of carotenoid accumulation in mature wolfberry fruit, a Pearson correlation analysis was performed based on estimates at the mature (S5) stages. First, the carotenoid content was analyzed by HPLC at five developmental stages in *L. barbarum* var. auranticarpum (Figure 7a). During fruit development, carotenoid contents (zeaxanthin, antheraxanthin, β-cryptoxanthin, and lutein palmitate) increased sharply as maturation progressed (Figure 7b). Among the four types of carotenoid metabolites, zeaxanthin was the most abundant in all stages of fruit development. Correlation tests were performed to evaluate relationships between abundances of various carotenoids (zeaxanthin, antheraxanthin, β-cryptoxanthin, and lutein palmitate) and transcript abundances of *LbaBBX* genes. As shown in Figure 7c, we observed a positive correlation between *LbaBBX25* with zeaxanthin (*r* = 0.967, *p* < 0.05) (Appendix A). Strong positive correlations were observed between transcript levels of *LbaBBX1* and *LbaBBX2* with antheraxanthin contents (*r* = 0.993, *p* < 0.001; *r* = 0.989, *p* < 0.001, respectively). Significant correlations were also observed between *LbaBBX11* (*r* = 0.985), *LbaBBX16* (*r* = 0.981), and *LbaBBX25* (*r* = 0.985) expression levels and antheraxanthin accumulation (all *p* < 0.05). Significant correlations were also observed between *LbaBBX11*, *LbaBBX16,* and *LbaBBX25* expression levels and antheraxanthin accumulation (*r* = 0.985, *p* < 0.05; *r* = 0.981, *p* < 0.05; *r* = 0.985, *p* < 0.05, respectively). However, weak negative correlations of *LaBBX9*, *LbaBBX12,* and *LbaBBX13* expression levels with antheraxanthin contents were observed. These correlations indicate that carotenoid accumulation in wolfberry fruits is correlated with the expression patterns of *LbaBBXs*.

To further investigate the regulation of *LbaBBX* genes in wolfberry, a correlation network was constructed combining four metabolites, 14 structural genes, and 13 LbaBBX TFs related to carotenoid biosynthesis. Only the pairs with a Pearson correlation coefficient >0.8 were included in this analysis (Figure 8). The network (visualized using Cytoscape) included 31 nodes connected by 123 edges. The pairwise correlations between genes (FPKM values) and between gene and metabolite levels revealed that 74 and 49 pairs of nodes, respectively, showed positive and negative correlations. As shown in Figure 8, all nine carotenoid biosynthesis genes exhibited positive correlations with carotenoid contents, with *LbaCYP97A29* showing the highest positive correlation (Appendix A). For the 13 LbaBBX TFs, the transcript changes in *LbaBBX1*, *LbaBBX2*, *LbaBBX4*, *LbaBBX11*, *LbaBBX16*, *LbaBBX18*, and *LbaBBX25* showed positive correlations, while *LbaBBX9*, *LbaBBX12*, and *LbaBBX13* showed negative correlations (Appendix A). For relationships between levels of carotenoid biosynthesis genes and BBX TFs, the highest positive correlation was observed between *LbaBBX2* and *LbaPDS*, followed by *LbaBBX1* and *LbaCRTISO*, while the highest negative correlation was found between LbaBBX11 and *LbaLCYE* (Appendix A). It is worth noting that *LbaBBX1*, *LbaBBX2*, *LbaBBX11*, and *LbaBBX16* levels showed strong positive correlations with levels of nine carotenoid biosynthesis genes each (Appendix A). These results indicated that these five *LbaBBXs* (*LbaBBX1*, *LbaBBX2*, *LbaBBX4*, *LbaBBX11*, and *LbaBBX16*) might be involved in the regulation of carotenoid biosynthesis.

### 2.9. Gene Expression Analyses by qRT-PCR

Nine potential *LbaBBXs* that showed strong correlations with the carotenoid content during fruit development we further evaluated by qRT-PCR. The expression patterns of several individual genes were highly correlated with the carotenoid content during wolfberry fruit development. Our results indicated that the expression levels of *LbaBBX2* and *LbaBBX4* increased sharply from S1 (12 DAF) to S3 (25 DAF) and reached peak values (Figure 9). The trends in the expression levels of these genes were consistent with trends in zeaxanthin content. Taken together, *LbaBBX2* and *LbaBBX4* were identified as important candidate genes involved in carotenoid biosynthesis and should be the focus of further functional research.

### 2.10. Subcellular Localization of LbaBBX2 and LbaBBX4

The candidate carotenoid-related genes *LbaBBX2* and *LbaBBX4* were selected for further analyses of subcellular localization. Their proteins were predicted to be located in the nucleus. To observe the subcellular localization of *LbaBBX2* and *LbaBBX4*, 35S-*LbaBBX2*::GFP and 35S-*LbaBBX4*:GFP were constructed and transiently expressed in tobacco leaves, and 35S-GFP was used as a negative control. As determined by fluorescence microscopy, the 35S-*LbaBBX2*::GFP and 35S-*LbaBBX4*::GFP fusion proteins were located exclusively in the nucleus, whereas the 35S-GFP control was distributed in the tobacco leaf protoplasts (Figure 10). These results indicate that both *LbaBBX2* and *LbaBBX4* encode nuclear-localized proteins.

## 3. Discussion

The *BBX* gene family has recently been identified in many higher plants, such as *Arabidopsis*, tomato, and pepper [7,25,26]. The quantity of *BBX* genes varies among species. For example, 32 *BBX* genes were identified in *Arabidopsis* [7], 31 in tomato [25], 24 in pepper [26], and 30 in potato [33]. In this study, 28 *LbaBBX* genes were identified in the wolfberry genome. The number of *BBX* genes in the four species in the family Solanaceae (tomato, pepper, potato, and wolfberry) was relatively conserved. However, there are 64 BBX members in apple [34]. Of note, the wolfberry genome (1.67 Gb) [31] is larger than those of *Arabidopsis* (134 Mb) [35] and tomato (900 Mb) [36], although it was smaller than the pepper genome (3.48 Gb) [37]. These results indicated that the number of BBX gene family members might not be directly related to the plant genome size. Furthermore, the composition of the *BBX* genes in different subclades also differed among species (Appendix A). In wolfberry, the numbers of BBX members with two tandem B-boxes plus the CCT domain, two tandem B-boxes, box 1 plus CCT, and B-box 1 were only 7, 9, 6, and 6, respectively. The corresponding counts were 13, 8, 4, and 7 in *Arabidopsis* [30] and 8, 11, 5, and 7 in tomatoes [25]. These results suggested that *BBX* genes shared a common ancestor and underwent an independent expansion after the divergence between monocots and dicots [38].

Previous phylogenetic analyses have verified that most plant *BBX* genes can be divided into five subgroups (Ⅰ–IV) [4,38]. In this present study, a phylogenetic tree based on BBX protein sequences from *Arabidopsis*, tomato, pepper, potato, eggplant, *Iochroma cyaneum**,* and wolfberry also supported their clustering into five subfamilies (Appendix A), consistent with previous results [38]. On the other hand, BBX proteins were grouped into five groups based on structure, depending on the presence of at least one B-box domain and a CCT domain. For example, 32 *Arabidopsis* BBXs were divided into five subclades according to a combination of conserved domains. The conserved domain-based classification of BBX proteins in *L. barbarum* was rather complex. As shown in Table 1, LbaBBX17, LbaBBX19, and LbaBBX21 were classified into group I, which had two B-boxes and a CCT plus a VP domain. Eight BBX members were classified into group II, including three LbaBBXs (LbaBBX9, LbaBBX10, and LbaBBX12) with one B-box plus a CCT domain, four LbaBBXs (LbaBBX8, LbaBBX13, LbaBBX20, and LbaBBX27) with two B-boxes and a CCT domain, and one LbaBBX (LbaBBX23) with two B-boxes. Group five contained only one B-box. A sequence alignment of LbaBBXs revealed a high degree of conservation of the B-Box1 domain among LbaBBX1 to LbaBBX28 (Appendix A). Thus, the clustering results were similar to those based on B-box 1. These results revealed that some LbaBBX proteins lost the B-box2 domain during evolution.

Gene duplication is one of the key factors responsible for the generation of novel genes, including WGD, TD, PD, TRD, and DSD, contributing to the expansion of gene family members in many species [32]. WGD, TD, and DSD are the main events in eukaryotic genome evolution and drove the development of new functions and genetic evolutionary systems [39]. Gene families, such as *R2R3-MYB* and *BAHD* acyltransferase families, expanded primarily through WGD and DSD [24,40]. WGD and TD are the main duplication events in the *PMEI* gene family [41]. In this study, DSD and WGD were the main factors driving *BBX* expansion in wolfberry, with relatively minor contributions from other replication modes.

The diversity of the biochemical functions of *BBX* genes has been identified in different species, including roles in plant photomorphogenesis, growth, development, metabolism, and responses to biotic or abiotic stresses [30]. For example, a number of BBXs, such as AtBBX21, AtBBX22, and AtBBX25, are involved in photomorphogenesis in *Arabidopsis* [42]. In apple, MdBBX37 is a negative regulator of anthocyanin accumulation via light signaling [43]. In tomatoes, *SlBBX17* is a positive regulator of heat stress tolerance [12]. Despite the diverse functions of *BBX* gene family members, we focus on their roles in carotenoid biosynthesis. Few studies have reported that *BBX* genes are involved in the regulation of carotenoid metabolism. In tomatoes, *SlBBX20* enhances carotenoid accumulation by activating *SlPSY1* promoter activity [11]. The 28 *LbaBBX* genes identified in this study showed variation in expression levels across the five stages of wolfberry fruit development. Based on transcriptome expression profiles combined with a correlation network analysis, expression levels of two candidate genes (*LbaBBX2* and *LbaBBX4*) belonging to clade Ⅳ were strongly correlated with carotenoid contents during fruit ripening. qRT-PCR analyses of *LbaBBX2* and *LbaBBX4* expression yielded consistent results, supporting the validity of the RNA-seq data. In addition, phylogenetic analyses indicated that LbaBBX2 and LbaBBX4 share high protein sequence homology with SlBBX20 (Appendix A). Together, we speculate that *LbaBBX2* and *LbaBBX4* are involved in the regulation of carotenoid biosynthesis in wolfberry. A limitation of this study is that a genetic transformation system for wolfberry plants has not been established. Therefore, the mechanism underlying *LbaBBX* gene expression dynamics in wolfberry plants still needs to be fully elucidated.

## 4. Materials and Methods

### 4.1. Plant Materials

Fruits of *Lycium barbarum* var. *auranticarpum* (with yellow fruit) were picked from the wolfberry germplasm of the National Wolfberry Engineering Research Center located at Yinchuan in Ningxia, China (38°38′49″ N, 106°9′10″ E). Fruit samples were harvested at five developmental stages (12, 19, 25, 30, and 37 days after full bloom, DAF). These fruits were immediately ground in liquid nitrogen and stored at −80 °C until further analysis.

### 4.2. Identification and Characterization of LbaBBX Genes in the L. barbarum Genome

In order to identify all possible BBX TFs in wolfberry, two strategies were used. In the first strategy, 31 *BBX* genes in the *Arabidopsis thaliana* genome were downloaded from the *Arabidopsis* Information Resource (TAIR, https://www.arabidopsis.org/, accessed on 4 January 2022) and were used as queries to search for potential *BBXs* in the *L. barbarum* genome database [31] by BLSATP with e value cutoff set at 1 × 10^−5^. In the second strategy, the HMMs of the B-box domain (PF00643) were downloaded from the Pfam database (https://pfam.xfam.org/, accessed on 8 January 2022), and HMMER 3.2 was used to identify BBX genes from the BLASTP alignments with default parameters. Subsequently, the presence of the BBX domain in each of the putative gene family members was further verified using the Pfam database (https://pfam.xfam.org/, accessed on 8 January 2022) [44], SMART database [45] (http://smart.embl-heidelberg.de/, accessed on 8 January 2022), and Conserved Domain Database (https://www.ncbi.nlm.nih.gov/cdd, accessed on 8 January 2022) [46]. Genes encoding proteins containing B-box domains were identified as BBX genes. The chemical properties, including the number of amino acids (aa), isoelectric point (pI), molecular weight (MW), and grand average of hydropathicity (GRAVY), were obtained from the ExPasy website (https://web.expasy.org/protparam/, accessed on 15 January 2022) [47]. The subcellular localizations of *LbaBBX* genes were predicted using WoLF PSORT (https://www.genscript.com/wolf-psort.html?src=leftbar, accessed on 15 January 2022) [48].

### 4.3. Multiple Sequence Alignment and Phylogenetic Analysis of LbaBBX Proteins

The BBX sequences for five species in Solanaceae, including tomato, pepper, eggplant, potato, and *Iochroma cyaneum,* were obtained from the Solanaceae Genomics Network (https://solgenomics.net/, accessed on 20 January 2022). First, full-length BBX protein sequences for the six species in Solanaceae and *Arabidopsis* were aligned by using Muscle v3.8 [49]. The deduced amino acid sequences in the B-box1, B-box2, and CCT domains were then adjusted manually using GeneDoc [50]. IQ-TREE [51] was used to construct a maximum likelihood (ML) phylogenetic tree based on all 194 full-length protein sequences. The best-fit substitution model, JTT+G, was determined using MEGA 6.06 [52]. The number of bootstrap replicates was 1000. The phylogenetic trees were visualized using iTOL v5 (https://itol.embl.de/, accessed on 20 January 2022) [53].

### 4.4. Gene Structure and Motif Composition of the LbaBBX Gene Family

The BBX genomic sequence and corresponding coding regions retrieved from the wolfberry genome were sent to the Gene Structure Display Server (http://gsds.gao-lab.org/, accessed on 25 January 2022) [54] to investigate exon–intron structures. MEME (https://meme-suite.org/meme/tools/meme, accessed on 25 January 2022) [55] was used to predict conserved motifs with a maximum number of motifs of 15 and optimum width of 3 to 50 bp.

### 4.5. Chromosomal Location and Gene Duplication Analysis of LbaBBX Genes

The chromosomal distribution of *LbaBBX* genes was visualized using TBtools [56]. To examine duplication events for *LbaBBX* genes in wolfberry and other plants, including *A. thaliana*, *S. lycopersicum*, and *C. annuum*, TBtools were used. The gene duplication pairs were visualized in Tbtools [56]. The whole-genome sequences of three species in Solanaceae, including *L. barbarum*, *S. lycopersicum*, *C. annuum,* and *A. thaliana*, were used to analyze collinearity. The detected syntenic blocks were visualized using Tbtools [56]. Furthermore, *K*_a_ and *K*_s_ substitution rates were calculated for each syntenic pair using KaKs_Calculator 3.0 [57].

### 4.6. Cis-Regulatory Elements in the Promoters of LbaBBX Genes

The 1500 bp genomic DNA sequences upstream of the start codon (ATG) of *LbaBBX* genes were extracted from the wolfberry genome database using Tbtools [56]. The *cis*-regulatory elements in these *LbaBBX* gene promoters were predicted by using PlantCARE (https://bioinformatics.psb.ugent.be/webtools/plantcare/html/, accessed on 3 March 2022).

### 4.7. RNA Isolation, cDNA Library Construction, and RNA-Seq Analysis

Total RNA was extracted independently from different wolfberry tissues using an RNA Kit (Tiangen, Beijing, China), according to the manufacturer’s instructions. RNA purity, concentration, and integrity were measured using a Nanodrop spectrophotometer (Thermo Scientific, Waltham, MA, USA) and an Agilent 2100 Bioanalyzer (Agilent Technologies, Santa Clara, CA, USA). High-quality RNAs were used to construct a cDNA library. First- and second-strand cDNAs were synthesized using Superscript II reverse transcriptase and random hexamer primers. Double-strand cDNA was fragmented by nebulization and used to generate RNA-seq libraries, as described previously [58]. Three biological replicates of cDNA libraries were sequenced using the Illumina HiSeq 4000 platform (Illumina Inc., San Diego, CA, USA) with a paired-end read length of 150 bp.

### 4.8. Expression Profiles of LbaBBXs

In order to study the expression profiles of *LbaBBX* genes, RNA-Seq data were downloaded from the NCBI database (PRJNA845109), including data for various tissues (stems, leaves, flowers, and fruits). The estimated expression levels of the *BBX* genes were represented and normalized in the form of fragments per kilobase of transcript per million mapped (FPKM). The heatmap for *LbaBBX* genes was visualized using Tbtools [56].

### 4.9. Quantitative Real-Time PCR (qRT-PCR) Analysis

RNA extraction and qRT-PCR were performed as previously described [24]. The primers for *LbaBBX* genes were designed using Primer Premier 5 and are listed in Appendix A. The wolfberry Actin gene was used as an internal control [20]. Three independent biological replications were conducted.

### 4.10. Carotenoid Extraction and HPLC Analysis

The extraction steps were as follows. Freeze-dried fruits were homogenized (30 Hz, 1.5 min) to a powder with a grinder (MM 400; Retsch, Haan, Germany). A mixture of *n*-hexane: acetone: ethanol (1:1:1, *v*/*v*/*v*) was prepared as an extraction solution, and then 0.01% BHT (g/mL) and 50 mg of power were mixed with an appropriate amount of extraction solution and internal standard. The extract was vortexed for 20 min at room temperature. The mixture was then centrifuged at 12,000 rpm/min for 5 min at 4 °C, and the supernatant was removed. The residue was re-extracted by repeating Steps under the same conditions. The supernatants were combined and evaporated to dryness. A mixture of methanol and methyl *tert* butyl ether was prepared; the sample was resuspended with an appropriate amount of the solution, vortexed thoroughly until it was fully dissolved, and centrifuged. The solution was filtered through a 0.22 μm membrane filter for further LC-MS/MS analysis [59]. Carotenoid contents were detected using the AB Sciex WTRAP 6500 LC-MS/MS platform by MetWare (Wuhan, China).

### 4.11. Correlation Network Construction

Expression patterns were explored based on RNA-seq data for five stages. The correlation coefficients for relationships between gene pairs and the carotenoid content were measured based on Pearson’s correlation coefficients (PCC). These values were screened using Excel (threshold > 0.8). A network including carotenoid contents, LbaBBX TFs, and structural genes was constructed and visualized using Cytoscape [60].

### 4.12. Subcellular Localization

For subcellular location assays, the 35S-*LbaBBX2::GFP*, *35S-LbaBBX4::GFP*, and 35S::*NLS-RFP* (control) constructs were introduced into tobacco (*Nicotiana benthamiana*) epidermal cells via *Agrobacterium* infiltrated tobacco leaves. Samples transformed with *35-GFP* were used as controls. After 2 days, GFP and RFP signals from the tobacco leaves inoculated with *A. tumefaciens* were detected by fluorescence microscopy (Olympus, BX63; Tokyo, Japan). Three independent experiments were performed for each gene.

### 4.13. Statistical Analyses

The data are presented as means ± SD of at least three independent experiments. Differences were evaluated by the Student’s *t*-test (* *p* < 0.05, ** *p* < 0.01, *** *p* < 0.001).

## 5. Conclusions

Our study provided the first genome-wide analysis of the *BBX* gene family in *L. barbarum*. A total of 28 *LbaBBXs* were identified and were unevenly distributed across the whole genome. A systematic and comprehensive analysis of the *LbaBBX* gene family was performed, including analyses of phylogenetic relationships, conserved domains, gene structure, motif composition, chromosome location, gene duplication, *cis*-acting elements, and expression patterns. Many *cis*-acting elements were found in the LbaBBX promoter sequences, indicating that *LbaBBX* genes are involved in complex regulatory networks controlling development. Correlation and qRT-PCR analyses revealed that *LbaBBX* genes might be involved in the regulation of carotenoid synthesis. Therefore, our genome-wide analysis of the BBX family provides a foundation for further studies of the molecular mechanisms underlying carotenoid synthesis in wolfberry.

## Figures and Tables

**Figure 1 ijms-23-08440-f001:**
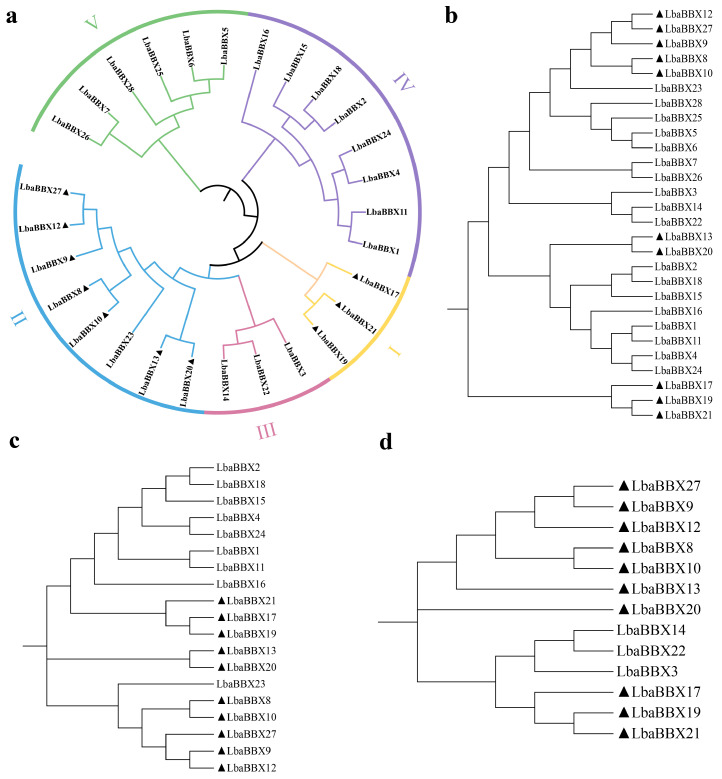
Phylogenetic tree analysis of BBX proteins in wolfberry. (**a**) The trees shown were based on the alignments of the protein sequences of the full length, and the phylogenetic tree was constructed using maximum likelihood method with 1000 bootstrap replicates by IQ-TREE. (**b**–**d**) The tree shown were based on the alignments of the protein’s sequences of the B-box 1 domain, B-box 2 domain and CCT domain, respectively. The members marked in black triangle contain two B-Box and one CCT domains.

**Figure 2 ijms-23-08440-f002:**
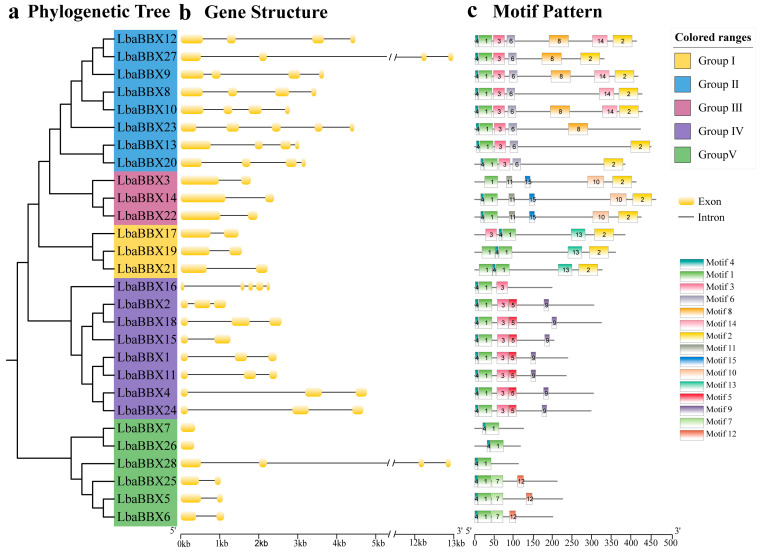
Phylogenetic relationships and motif composition of the LbaBBX proteins, and gene structure of the *LbaBBX* genes. (**a**) The phylogenetic tree was constructed based on the full-length sequences using IQ-TREE software by maximum likelihood (ML) method and 1000 bootstrap replicates. (**b**) Exon/intron structures of *BBX* genes from wolfberry. The exons and introns are represented by yellow boxes and black lines, respectively. The sizes of exons and introns can be estimated using the scale below. (**c**) The conserved motifs of wolfberry proteins were elucidated by MEME. The 15 motifs were displayed by the different colored rectangles. The sequence information for each motif is provided in Appendix A. The length of protein can be estimated using the scale at the bottom.

**Figure 3 ijms-23-08440-f003:**
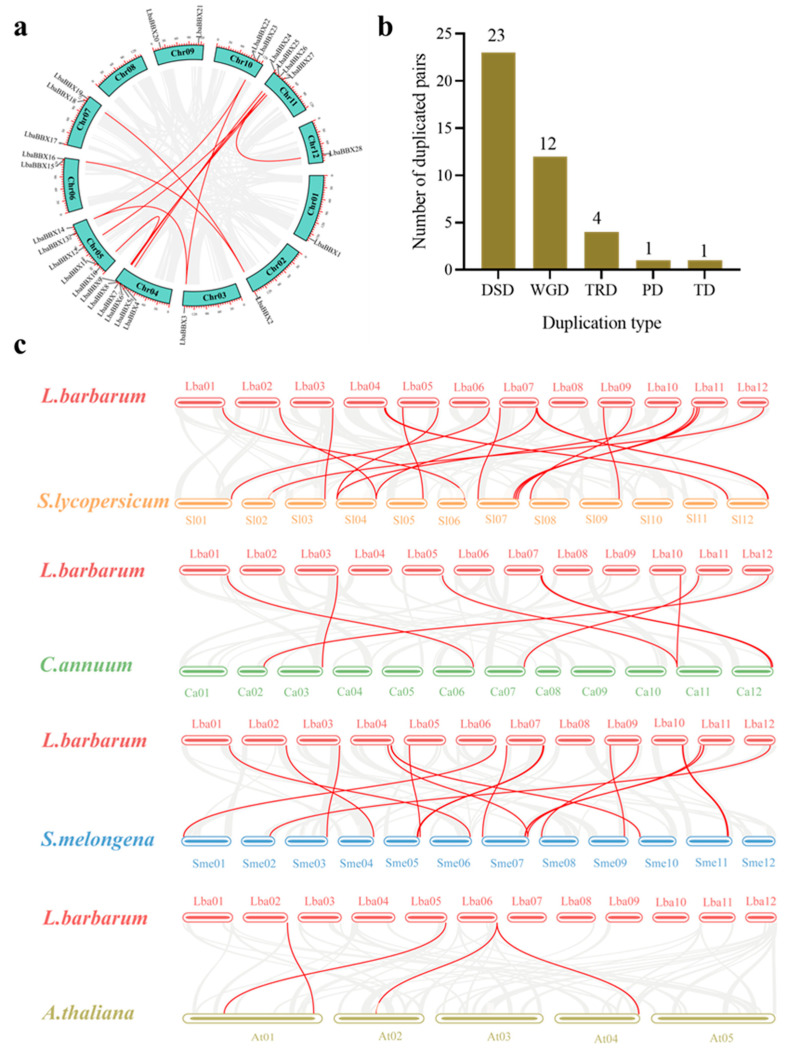
Chromosomal location and duplicated genes among *LbaBBX* genes. (**a**) Intraspecific collinearity analysis. A total of 28 *LbaBBX* genes were mapped onto the chromosomes based on their physical location. The red lines indicate duplicated *LbaBBX* gene pairs. (**b**) Different models of gene duplication in *LbaBBX* family. The *x*-axis represents the duplication type. The *y*-axis represents the number of duplicated gene pairs. Whole genome duplication (WGD), tandem duplication (TD), proximal duplication (PD), transposed duplication (TRD), dispersed duplication (DSD). (**c**) Analysis of collinearity between two different species. The gray lines indicated duplicated blocks, while the red lines indicated the syntenic *BBX* gene pairs. Chromosome numbers are indicated above or below each chromosome.

**Figure 4 ijms-23-08440-f004:**
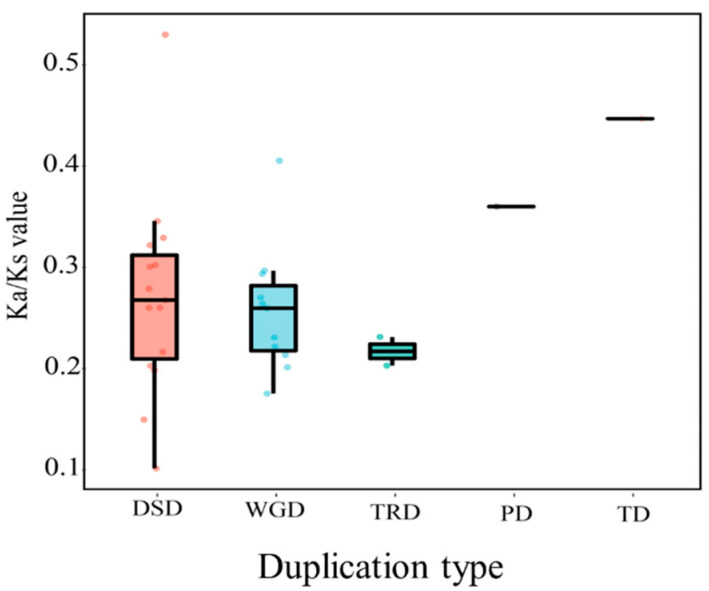
Different models of gene duplication in the *LbaBBX* family. The *x*-axis represents the duplication type. The colorful dots represent duplicated gene pairs. Whole genome duplication (WGD), tandem duplication (TD), proximal duplication (PD), transposed duplication (TRD), and dispersed duplication (DSD).

**Figure 5 ijms-23-08440-f005:**
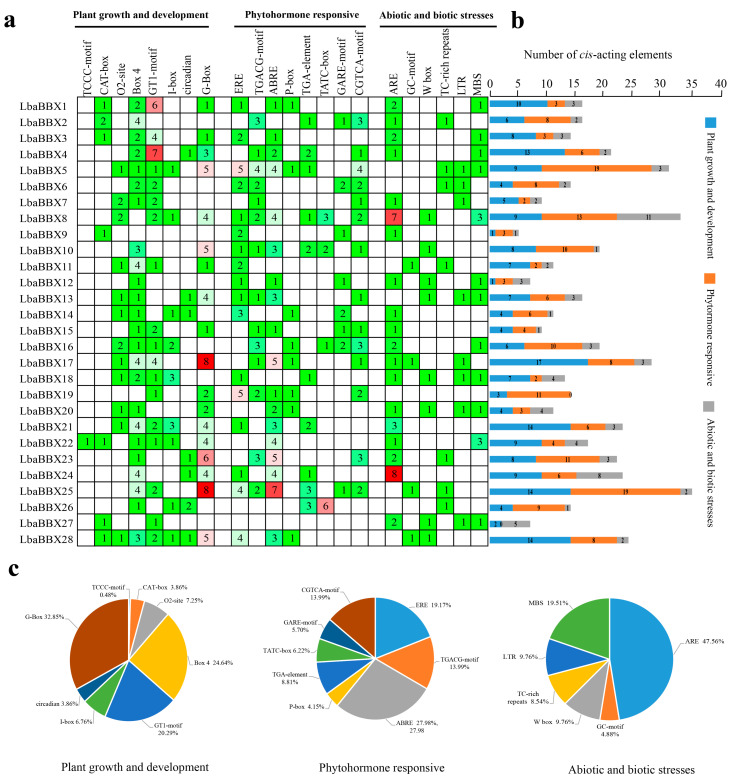
Identification of *cis*-elements in promoter regions of *LbaBBXs*. (**a**) Three categories of *cis*-acting elements in the *LbaBBXs.* Different numbers and color of the gird representing the number of different elements in these *LbaBBXs*. (**b**) Histogram of the cis-acting elements in each *LbaBBX* gene. The blue rectangle represents plant growth and development responsive *cis*-elements, the orange rectangle represents phytohormone responsive *cis*-elements, and the gray rectangle represents abiotic and biotic stress responsive cis-elements. (**c**) Pie charts of different sizes indicated the ration of each promoter element in each category, respectively.

**Figure 6 ijms-23-08440-f006:**
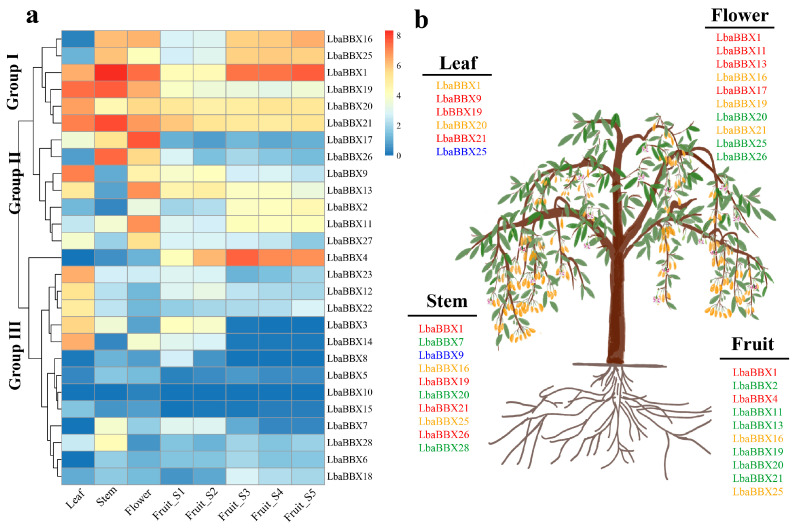
Expression pattern of *LbaBBX* genes. (**a**) Tissue-specific expression pattern of LbaBBX genes in four tissues: leaf, stem, flower, and fruits, including five development stages. Blue and red color indicated lower and higher transcript abundance, respectively. (**b**) Identification of highly expressed *BBX* genes in *L. barbarum*. Blue, green, orange, red indicated low (1–7.3 FPKM), mid-low (7.3–48 FPKM), mid-high (48–114 FPKM), and high (114–317 FPKM) expression, respectively.

**Figure 7 ijms-23-08440-f007:**
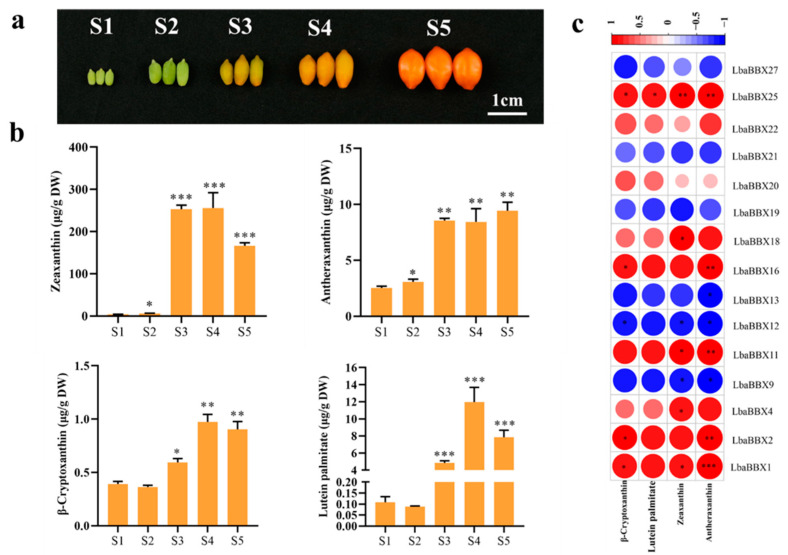
Identification *LbaBBX* genes related to carotenoid biosynthesis. (**a**) Fruits of *Lycium barbarum* var. auranticarpum at different stages of development. S1, S2, S3, S4, and S5 period represent 12, 19, 25, 30, and 37 days after full bloom (DAF), respectively. Scale bars represent 1 cm. (**b**) Trends in carotenoids (zeaxanthin, antheraxanthin, β–cryptoxanthin and lutein palmitate) at five developmental stages. The data contain the averages and standard deviations of three individual replicates. Asterisks indicate a significant difference (* *p* < 0.05, ** *p* < 0.01, *** *p* < 0.001) compared with S1 at the different time points during development. (**c**) Correlation analysis was constructed using the expression levels of *LbaBBX* genes and carotenoids content in five different developmental stages. The blue color means negative correlation, the red color means positive correlation. Asterisks indicate a significant difference (* *p* < 0.05, ** *p* < 0.01, *** *p* < 0.001).

**Figure 8 ijms-23-08440-f008:**
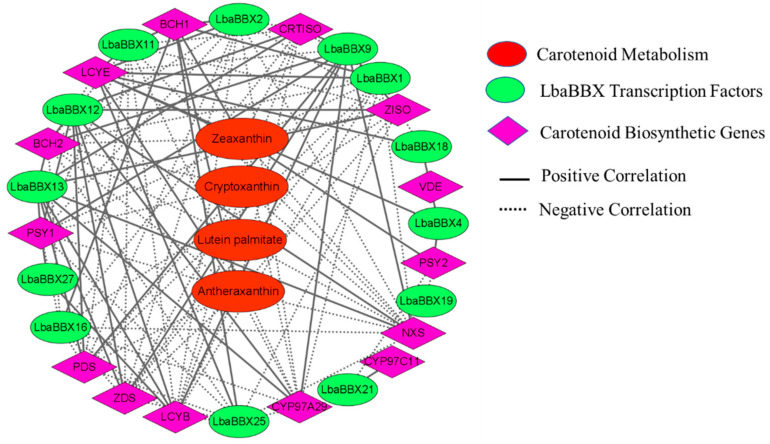
Correlation network analysis for structural genes, LbaBBX transcription factors and carotenoid content. The red ellipse boxes indicated that carotenoid metabolism, the green ellipse boxes indicated that LbaBBX transcription factors, and the purple diamond boxes indicated carotenoid biosynthetic genes, respectively. The black solid lines indicated positive regulation, while the dot lines indicated negative regulation, respectively. The edges are drawn when the linear correlation coefficient is >0.8 with *p*-value < 0.05. The related correlation coefficients were shown in Appendix A.

**Figure 9 ijms-23-08440-f009:**
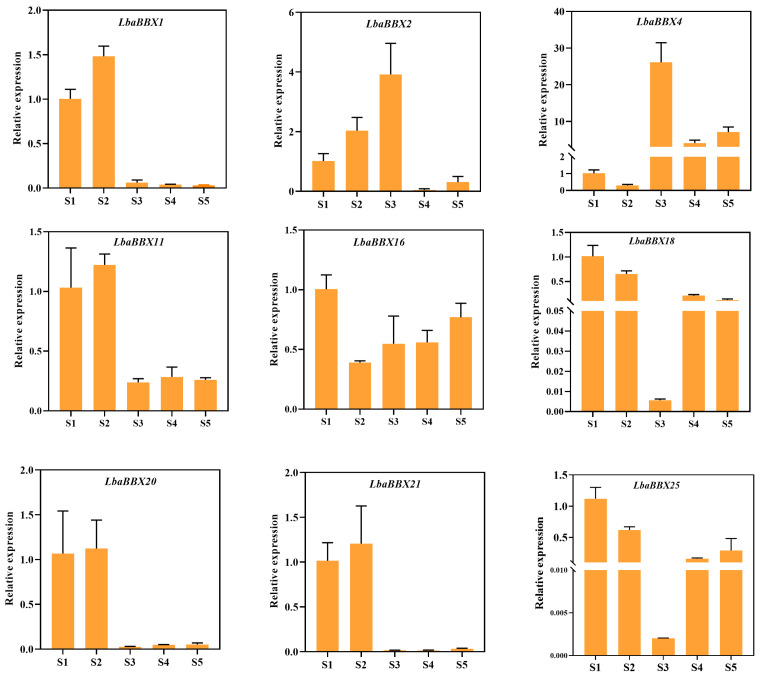
The relative expression levels of nine *LbaBBX* genes at different fruit developmental stages. Actin gene was used as reference gene to measure expression levels in each period. The *x*-axis indicates the five distinct fruit developmental stages (12 DAF, 19 DAF, 25 DAF, 30 DAF and 37 DAF). The *y*-axis indicates the relative expression. Data represent the means ± SDs (*n* = 3).

**Figure 10 ijms-23-08440-f010:**
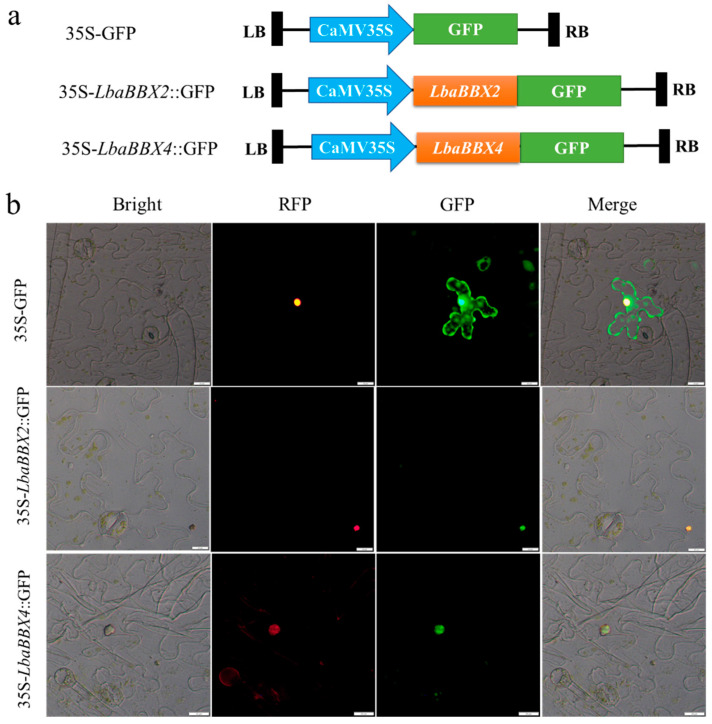
Subcellular localization of LbaBBX4 protein. (**a**) Schematic diagram of the 35S-GFP, 35S-*LbaBBX2*::GFP, and 35S-*LbaBBX4*::GFP fusion protein constructs used for transient expression. (**b**) The LbaBBX2-GFP and *LbaBBX4*-GFP fusion proteins were transiently expressed in *N. benthamiana* leaves and observed by fluorescence microscopy 48 h later. The 35S-GFP was used as positive control. From left to right, bright field, red fluorescent protein (RFP) (nuclear localization signal (NLS)-RFP), green fluorescent protein (GFP), and merge image of RFP and GFP image. Scale bars =20 μm.

**Table 1 ijms-23-08440-t001:** Information on the *BBX* gene family in wolfberry.

Gene ID	Gene Name	CDS	AA	pI	*M*_W_ (kDa)	GRAVY	Subcellular Localization	Domains	Structure
*Lba01g02500*	*LbaBBX1*	705	234	4.91	25.99	−0.393	nucleus	2BBX	Ⅳ
*Lba02g02688*	*LbaBBX2*	903	300	6.20	33.42	−0.522	nucleus	2BBX	Ⅳ
*Lba03g02797*	*LbaBBX3*	1224	407	5.23	46.60	−0.86	nucleus	1BBX + CCT	Ⅲ
*Lba04g02191*	*LbaBBX4*	900	299	5.13	32.15	−0.272	nucleus	2BBX	Ⅳ
*Lba04g02506*	*LbaBBX5*	666	221	5.03	24.38	−0.582	nucleus	1BBX	Ⅴ
*Lba04g02507*	*LbaBBX6*	591	196	5.10	21.65	−0.683	chloroplast	1BBX	Ⅴ
*Lba04g02527*	*LbaBBX7*	369	122	7.52	13.83	−0.241	nucleus	1BBX	Ⅴ
*Lba04g02528*	*LbaBBX8*	1269	422	6.08	46.01	−0.564	nucleus	2BBX + CCT	Ⅱ
*Lba04g02630*	*LbaBBX9*	1239	412	5.13	45.55	−0.656	nucleus	1BBX + CCT	Ⅱ
*Lba05g00735*	*LbaBBX10*	1272	423	5.24	46.57	−0.508	nucleus	1BBX + CCT	Ⅱ
*Lba05g00905*	*LbaBBX11*	693	230	5.63	25.67	−0.343	nucleus	2BBX	Ⅳ
*Lba05g01291*	*LbaBBX12*	1227	408	5.64	44.95	−0.601	nucleus	1BBX + CCT	Ⅱ
*Lba05g01679*	*LbaBBX13*	1338	445	7.05	48.87	−0.603	nucleus	2BBX + CCT	Ⅱ
*Lba05g02193*	*LbaBBX14*	1374	457	5.24	51.73	−0.741	nucleus	1BBX + CCT	Ⅲ
*Lba06g03364*	*LbaBBX15*	600	199	5.88	21.86	−0.518	nucleus	2BBX	Ⅳ
*Lba06g03380*	*LbaBBX16*	585	194	5.88	21.54	−0.573	chloroplast	2BBX	Ⅳ
*Lba07g00041*	*LbaBBX17*	1140	379	6.70	41.94	−0.35	chloroplast	2BBX + CCT	Ⅰ
*Lba07g01710*	*LbaBBX18*	960	319	8.40	35.23	−0.479	nucleus	2BBX	Ⅳ
*Lba07g01848*	*LbaBBX19*	1068	355	5.96	39.31	−0.561	chloroplast	2BBX + CCT	Ⅰ
*Lba09g00845*	*LbaBBX20*	1140	379	5.27	42.66	−0.728	nucleus	2BBX + CCT	Ⅱ
*Lba09g01983*	*LbaBBX21*	966	321	5.69	35.91	−0.592	cytoplasmic	2BBX + CCT	Ⅰ
*Lba10g01709*	*LbaBBX22*	1263	420	5.37	48.09	−0.824	nucleus	1BBX + CCT	Ⅲ
*Lba10g01753*	*LbaBBX23*	1257	418	4.94	46.75	−0.506	nucleus	2BBX	Ⅱ
*Lba11g00500*	*LbaBBX24*	882	293	4.98	31.56	−0.404	nucleus	2BBX	Ⅳ
*Lba11g00948*	*LbaBBX25*	624	207	4.71	23.43	−0.916	nucleus	1BBX	Ⅴ
*Lba11g00982*	*LbaBBX26*	345	114	8.56	12.96	−0.421	cytoplasmic	1BBX	Ⅴ
*Lba11g01258*	*LbaBBX27*	981	326	4.65	35.69	−0.423	nucleus	2BBX + CCT	Ⅱ
*Lba12g01725*	*LbaBBX28*	330	109	9.21	12.49	−0.446	cytoplasmic	1BBX	Ⅴ

## Data Availability

The wolfberry genome datasets used during the current study are available in NCBI database (https://www.ncbi.nlm.nih.gov/bioproject/PRJNA640228, accessed on 20 December 2021). The tomato, pepper, eggplant, and potato genome sequences were downloaded from the Genome Database for the Solanaceae (https://solgenomics.net/, accessed on 20 December 2021). The sequence of *Arabidopsis* was downloaded from the Arabidopsis Information Resource (https://www.arabidopsis.org/, accessed on 20 December 2021). The raw data of the transcriptome analysis used in this study were submitted to the Sequence Read Archive (SRA) at a NCBI database (PRJNA845109).

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
