# Peer review of "Genome-Wide Identification and Analysis of the BBX Gene Family and Its Role in Carotenoid Biosynthesis in Wolfberry (Lycium barbarum L.)"

_ijms, 2022, doi:10.3390/ijms23158440_

Round 1
Reviewer 1 Report
This manuscript describes the BBX transcription factor family in wolfberry. The manuscript has searched for BBX genes in wolfberry through genome sequence and performed a comparative genomic analysis with Arabidopsis thaliana and other species. Furthermore, the correlation with carotenoid accumulation is analyzed, and the possibility that two BBXs are involved in regulating carotenoid biosynthesis is proposed.
The data reported in this manuscript may contribute to elucidating the regulatory mechanisms of carotenoid production in wolfberry. However, the following minor points should be corrected or addressed.
1. The results of the correlation analysis do not provide direct evidence of the involvement of BBX in carotenoid biosynthesis, as described in lines 337 and 338. The statement "LbaBBX2 and LbaBBX4 played key roles in ~" in line 29 should be revised.
2. What does RFP show in subcellular localization analysis? Is it a positive control for nuclear localization? Please explain it in the methods and figure legend.
Author Response
Response to Reviewer 1 Comments
Point1: The results of the correlation analysis do not provide direct evidence of the involvement of BBX in carotenoid biosynthesis, as described in lines 337 and 338. The statement "LbaBBX2 and LbaBBX4 played key roles in ~" in line 29 should be revised.
Response 1: In line 29, the sentence revised as following:
LbaBBX2 and LbaBBX4 might played key roles in the regulation of zeaxanthin and antheraxanthin.
Point 2: What does RFP show in subcellular localization analysis? Is it a positive control for nuclear localization? Please explain it in the methods and figure legend
Response 2: RFP represent red fluorescent protein (nuclear localization signal (NLS)-RFP). It is a positive control for nuclear location.

Reviewer 2 Report
The paper provides a comprehensive analysis of the B-box protein genes in wolfberry (Lycium barbarum L.) concluding that LbaBBX genes are involved in complex regulatory networks and carotenoid synthesis might be one of them. Scientific evidence documented Lycium barbarum L., as an important medicinal plant and food supplement, cultivated for its fruits, known as goji berry. This study identified 28 LbaBBX genes and revealed these to be highly responsive to light, phytohormone, and stress conditions. A synteny analysis indicated that several LbaBBX genes were orthologous to three other plants in Solanaceae family (Solanum lycopersicum, Solanum melongena, Capsicum annuum) and one plant in Brassicaceae ( Arabidopsis thaliana). Because wolfberry is rich in carotenoids, their content were assessed, and carotenoid gene expression patterns and qRT-PCR analyses, identifying the genes that might be involved in the biosynthesis of zeaxanthin and antheraxanthin. The subject is important, the manuscript is well written, and, given the complexity involved, the authors made significant contributions. In summary, the title stresses the value of the study; the abstract includes sufficient information to stand alone and is clearly structured on the IMRAD format; the introduction summarizes the topic current state and knowledge in the field and explains why the experiment was needed; the results are accurately presented, with relevant data given in tables and figures; in the discussion chapter the findings of the study are logically explained and compared to other findings in the field, followed by limitations of the study; the methods are distinctly described with adequate details; the conclusions support the genome-wide analysis of the BBX family and indicate potential future research addressing the mechanisms responsible for carotenoid synthesis in wolfberry.
I have no hesitation in recommending this manuscript for publication after a few typos and minor details have been attended to:
Pag 5, Fig 1 (Legend): “The trees shown were...”
Pag 6, Fig 2 (top): “Phylogenetic Tree”
Pag 7, First paragraph: “were greater”; “tomato”
Pag 8, First line: “highlight” can be deleted
Pag 14, Lines 163-164: ”were only” instead of “only were”
Pag 15, Line 211: second “high” can be deleted
Pag 16, Line 221: “at” instead of “from”
Author Response
Point 1: Pag 5, Fig 1 (Legend): “The trees shown were...”
Response 1: The sentence is revised. (line 149, red marker)
Point 2: Pag 6, Fig 2 (top): “Phylogenetic Tree”
Response 2: The word was rerevised.
Point 2: Pag 7, First paragraph: “were greater”; “tomato”
Response 2: Theses words are revised.
Point 3 : Pag 8, First line: “highlight” can be deleted
Response 3: The “highlight” can be deleted”
Point 4: Pag 14, Lines 163-164: ”were only” instead of “only were”
Response 4: The words were revised.
Point 5: Pag 15, Line 211: second “high” can be deleted
Response 5: The second “high” can be deleted.
Point 5: Pag 16, Line 221: “at” instead of “from”
Response 6: The words were revised.
